# Layer-selective hydrogenation and proton transport in twisted bilayer graphene

J. Tong [1,2] ✉, G. Chen[1,2], H. Li[1], E. Hoenig[1,2], M. Alhashmi[1,2], X. Zhang[1,2], D. Bahamon[3,4], G. R. Tainton [2,5], S. Sullivan-Allsop [2,5], Y. Mayamei [1,2], D. R. da Costa [6], L. F. Vega [3,4], S. J. Haigh [2,5], D. Domaretskiy [1], F. M. Peeters [6,7,8] & M. Lozada-Hidalgo [1,2] ✉

Recent work investigated graphene's hydrogenation with independent control of the electric field, $E$, and charge density, $n$, in the crystal and showed that the process is controlled by $n$. Here, we demonstrate layer-selective conductor–insulator transitions in twisted bilayer graphene, driven by hydrogenation at fixed $n$ under strong $E$. This process is accompanied by proton transport through the bilayer, enabling several parallel and configurable logic gates in the devices. Selectivity arises because the large twist angle decouples the two layers' electronic systems, enabling independent control of their charge densities. Polarisation by the field then induces a charge imbalance at fixed total $n$, triggering hydrogenation when one of the layers' charge densities reaches the threshold for monolayer hydrogenation. Our results introduce a new type of electrode-electrolyte interface in which electrochemical processes are controlled with two decoupled 2D electron gases, opening new design opportunities for energy and information processing devices.

In a classical electrode-electrolyte interface, the applied potential couples the charge density, $n$, and the electric field perpendicular to the electrode, $E$, via parameters, such as solvent polarisability or Debye length[1,2]. In contrast, electrostatically gating a 2D crystal on both of its surfaces—a technique known as double gating—enables independent control of $E$ and $n$ in the crystal[3–8]. This decoupling expands the parameter space under experimental control, opening otherwise inaccessible pathways to drive electrochemical processes[2]. In recent work[2], we applied this approach to a graphene electrochemical interface and demonstrated that it enables selective control of two otherwise coupled processes: proton transport through the crystal's basal plane[9–12] and proton chemisorption (hydrogenation)[13–16]. Proton transport was driven by $E$, and at high fields (~1 V nm⁻¹) graphene

became effectively transparent to protons. Conversely, hydrogenation was driven exclusively by $n$. This process led to a robust and reversible conductor-insulator transition in the crystal that, combined with the proton transport signal, enabled proton-based logic-and-memory operations in graphene[2]. Beyond electrochemical processes, double-gating has been routinely used to control diverse electronic phenomena in 2D heterostructures, such as ferroelectric switching[17,18] and bandgap quenching in 2D semiconductors[3,19], or to optically map the chemical potential of electrons in 2D heterostructures[8]. These advances could enable new ways of controlling electrochemical processes via the field-tuneable electronic properties of 2D systems. In this work, we studied proton transport and hydrogenation in non-aligned bilayer graphene. Although the two graphene layers are in direct contact, the

[1]Department of Physics and Astronomy, The University of Manchester, Manchester, UK. [2]National Graphene Institute, The University of Manchester, Manchester, UK. [3]Research and Innovation Center on CO2 and Hydrogen (RICH Center) and Chemical Engineering Department, Khalifa University, Abu Dhabi, UAE. [4]Research and Innovation Center for graphene and 2D materials (RIC2D), Khalifa University, Abu Dhabi, UAE. [5]Department of Materials, The University of Manchester, Manchester, UK. [6]Departamento de Física, Universidade Federal do Ceará, Fortaleza, Ceará, Brazil. [7]School of Physics and Optoelectronic Engineering, Nanjing University of Information Science and Technology, Nanjing, China. [8]Departement Fysica, Universiteit Antwerpen, Antwerp, Belgium. ✉e-mail: tongjincheng@outlook.com; marcelo.lozadahidalgo@manchester.ac.uk

large twist angle induces a momentum mismatch between electrons in each layer, which decouples their wavefunctions[20–22]. This constitutes a new type of electrode-electrolyte interface and remains unexplored.

## Results

The devices for this work were fabricated by suspending mechanically exfoliated, non-aligned bilayer graphene over a 10 μm hole etched in a silicon nitride (SiNx) substrate (Fig. 1 and Supplementary Figs. 1 and 2). The suspended membranes were coated on both sides with a non-aqueous, proton-conducting electrolyte−bis(trifluoromethane)sulfonimide (HTFSI) in polyethylene glycol−with a wide electrochemical stability window (>4 V). The devices were contacted with two palladium hydride (PdHx) electrodes and connected to the electrical circuit shown in Fig. 1b ('Device fabrication' in Methods). Two gate voltages, $V_t$

and $V_b$, were applied between the bilayer and the respective PdHx electrodes, enabling independent control of the potential at each graphene–electrolyte interface (Supplementary Fig. 3)[2,3]. We studied two electrochemical processes. The first was the electrochemical hydrogenation of twisted bilayer graphene. Graphene's hydrogenation is characterised by a reversible conductor-insulator transition in the crystal, accompanied by a D band in its Raman spectra[2,13,23]. To monitor this process, we measured the in-plane electronic conductivity of each layer independently as a function of $V_t$ and $V_b$, by applying in-plane source-drain voltages to each layer, $V_{tds}$ and $V_{bds}$. Alternatively, we measured the electron tunnelling current and voltage between the two layers. The second process was out-of-plane proton transport[2,9,10], which is characterised by the current in the electrolyte gates. We studied these processes with independent control of the electric field and

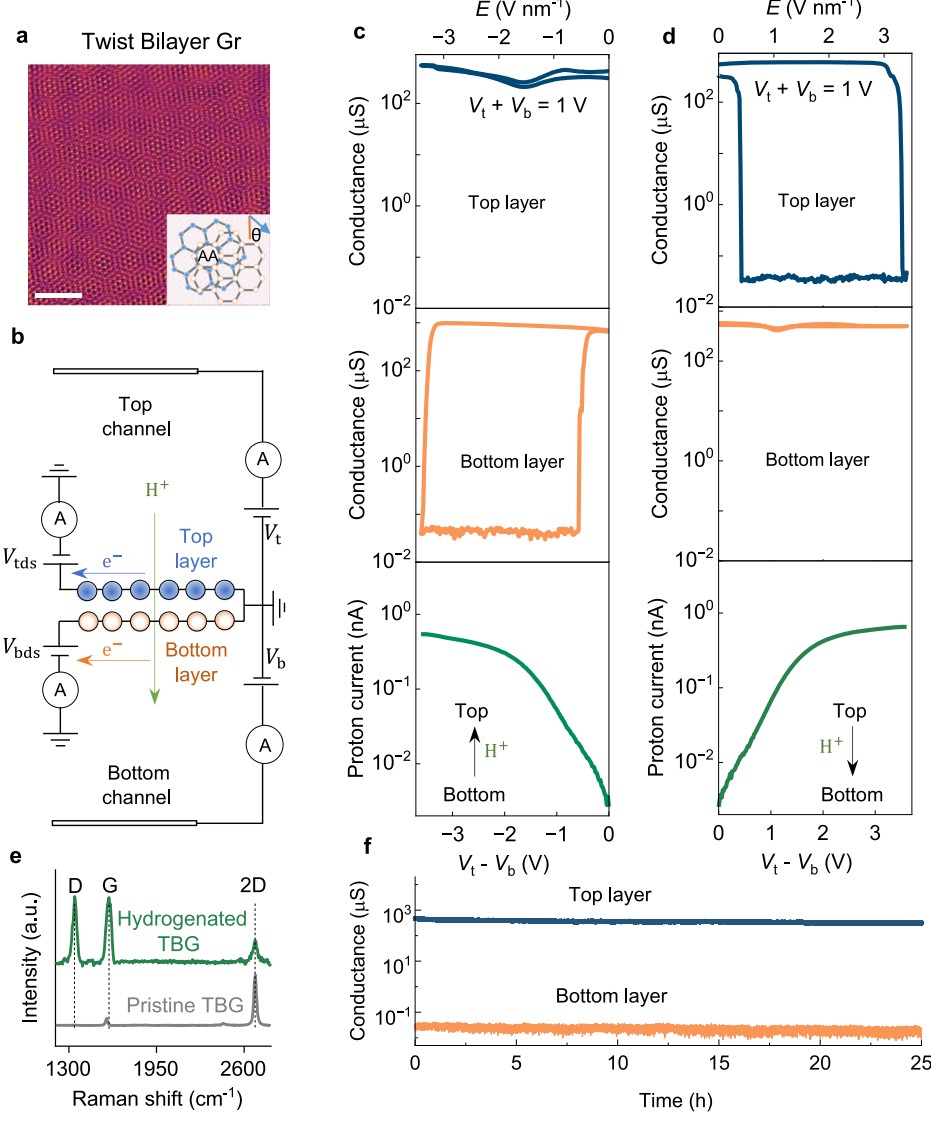

**Fig. 1 | Layer-selective hydrogenation and proton transport in twisted bilayer graphene. a** Atomic resolution high angle annular dark field scanning transmission electron microscopy image of a representative bilayer graphene stack with a twist angle (θ) of 12.5°. Scale bar, 2 nm. Inset, schematic of bilayer graphene with a twist angle. AA regions indicate atomic alignment between layers. **b** Schematic of double-gated configuration. **c** Top and middle panels: In-plane electronic conductance of top (blue) and bottom (orange) graphene layers under negative electric field ($V_t − V_b < 0$) at constant charge density ($V_t + V_b = 1$ V, corresponding to $n ≈ 5 × 10^{13}$ cm$^{-2}$). $n$ is below the threshold of ≈1×10$^{14}$ cm$^{-2}$ required to hydrogenate graphene found in ref. 2. The bottom layer shows a reversible conductor–insulator

transition, and the top layer remains conductive. $V_{tds} = V_{bds} = 2$ mV. Bottom panel, simultaneous measurement of out-of-plane proton current. The field drives protons from bottom to top channel. **d** Same measurements as in panel **c**, under positive electric field ($V_t − V_b > 0$). The bottom layer now remains conductive, while the top layer is hydrogenated. **e** Raman spectra before and after single-layer hydrogenation. The latter spectrum displays a prominent D band. **f** Single-layer hydrogenated state obtained after a hydrogenation cycle and then setting $E = −1.1$ V nm$^{-1}$ ($V_t − V_b = −1$ V), $n ≈ 0.5 × 10^{14}$ cm$^{-2}$. The top layer remains conductive and the bottom layer insulating for >24 h.

charge density in the double-gated crystal. As established in multiple works[3–8], the electric field is determined by the difference in the gate voltages, $E \propto V_t - V_b$, and the charge density, $n$, by their sum, $V_t + V_b$ ('Estimation of $E$ and $n$' in Methods).

We started by investigating these processes under much higher $E$ than in our previous work, while fixing the charge density at $n \approx 5 \times 10^{13}\,\text{cm}^{-2}$, which is below the threshold required to hydrogenate graphene found in Refs. 2,13 ($1 \times 10^{14}\,\text{cm}^{-2}$). Starting with proton transport, the bottom panel of Fig. 1c shows that the electric field drove protons through the bilayer, yielding currents about 100 times lower than in monolayer graphene[2] for the same $E$. This arises for two reasons. First, AB-stacked bilayers are impermeable to protons[2,9], so proton transport can be expected to take place through AA-stacked regions, which occupy only ≈30% of the crystal's area[24]. Second, AA-stacked regions in the bilayer pose a larger barrier to incoming protons than monolayer graphene (Supplementary Fig. 4). Unexpectedly, for a sufficiently high electric field, the layer facing incoming protons was hydrogenated, even though $n$ remained constant and below the threshold required to enable this process. The hydrogenation of this layer was evidenced by a conductor-insulator transition in its electronic response, accompanied by a prominent D band in the sample's Raman spectrum (Fig. 1c, e). Notably, only this layer was hydrogenated, and the resulting single-layer insulating state was stable for over 24 h under constant gate voltages, demonstrating robust state retention (Fig. 1f). Reducing $E$ below -0.5 V nm$^{-1}$ (at constant $n$), recovered the layer's conductivity, yielding a hysteretic electronic response and the disappearance of the Raman D band (Supplementary Fig. 5). The selective hydrogenation process was highly reproducible, displaying ON/OFF ratios of >$10^4$ with cycle-to-cycle variation of less than 6% for over 1000 cycles without degradation (Supplementary Fig. 6). The

hydrogenation process was symmetric with respect to the direction of the electric field. Reversing the polarity of the field at fixed $n$ reversed the direction of proton flow, and now the opposite layer was hydrogenated (Fig. 1d). These results demonstrate that a strong $E$ in twisted bilayer graphene can enable layer-selective hydrogenation even at $n$ previously considered too low to sustain this process.

We then characterised this process for various fixed $n$. To that end, we measured electron tunnelling between the two layers, which provides a four-probe, single-signal characterisation of the devices' electronic properties and, since hydrogenation of either layer suppresses tunnelling, it characterises the insulating transition (Fig. 2a). During the measurements, we fixed $n$ and swept $E$ in a single polarity ($E > 0$) to minimise exposure of the devices to extreme fields and ensure stability. Figure 2b shows that for $n < 5 \times 10^{13}\,\text{cm}^2$ ($V_t + V_b < 1$ V), no insulating transition was observed within the range of $E$ applied; whereas for $n = 5 \times 10^{13}\,\text{cm}^{-2}$, the system reached the insulating state at high $E$, as discussed above. However, as $n$ increased, we found that the field required to reach the insulating state decreased, and for large $n > 2 \times 10^{14}\,\text{cm}^{-2}$ ($V_t + V_b > 4$ V), it was reached even at $E = 0$. These observations show that lower charge density requires a stronger electric field to induce hydrogenation in this system.

To understand these findings, we measured the full $E$-$n$ map of the electronic response (Fig. 2d). This revealed that the device displayed two neutrality curves, rather than the single neutrality line in monolayer graphene[2] and AB-stacked bilayer graphene (Supplementary Fig. 7, 'AB stacked bilayers' in Methods). The splitting of the neutrality line has been reported previously and shown to arise from the polarisation of the crystal under the applied $E$[20]. However, in double-ionic-gated devices $E$ is much stronger than in devices with solid-state dielectrics, which strongly polarise the bilayer, and now yield a clear

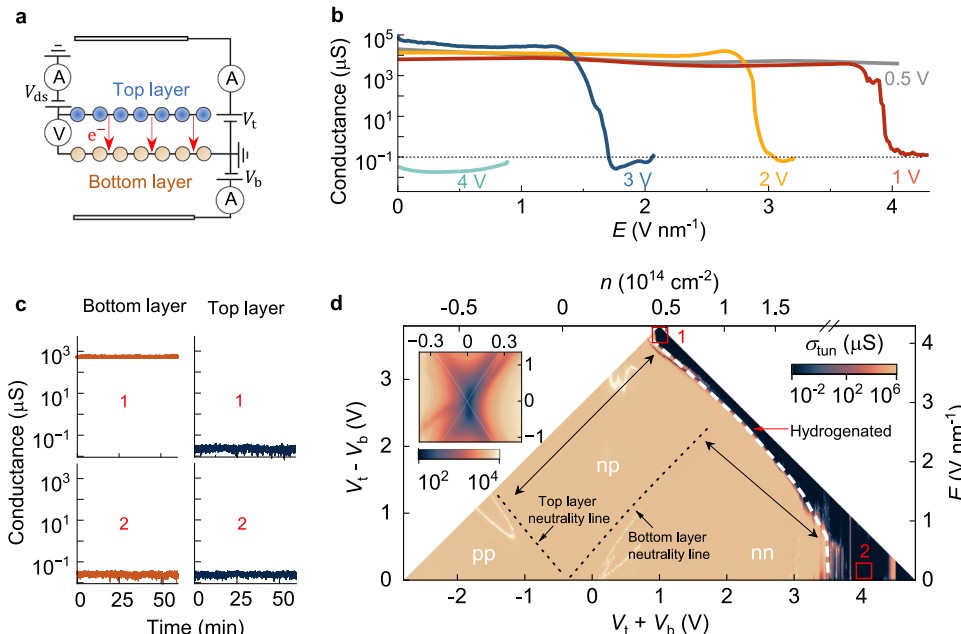

**Fig. 2 | Field effect control of layer-selective hydrogenation. a** Schematic of device and electronic circuit. Interlayer tunnelling current ($I_{tun}$, marked with red arrows) and tunnelling voltage ($V_{tun}$) between the graphene layers are measured simultaneously to obtain tunnelling conductance ($\sigma_{tun} = I_{tun}/V_{tun}$). **b** Tunnelling conductance as a function of $E$ at fixed $n$ (constant $V_t + V_b$ indicated). Dashed line, guide to the eye marking insulting state. **c** Time-resolved in-plane electronic conductance in each layer (same electronic circuit as that of Fig. 1) for the gate voltage combinations marked with red boxes in (**d**). In box 1 (top row), the bottom layer (red) is conducting, and the top (blue) is insulating; in box 2 (bottom row), both layers are insulating. **d** Map of tunnelling conductance as a function of electric field ($E$, right $y$-axis) and charge density ($n$, top $x$-axis). The left and bottom axes

correspond to the control voltages $V_t - V_b$ and $V_t + V_b$, respectively. Dashed black lines mark the neutrality lines from each layer. Black region in the map corresponds to insulating states. Arrows mark that the boundary of the insulating region runs along the neutrality lines from each layer. White labels (pp, np, and nn) mark the doping regime of the top and bottom layers. Red boxes indicate gate voltage combinations probed in time-resolved measurements in panel **c**. Inset, electronic response as a function of $n$ (top $x$-axis) and $E$ (right $y$-axis) for lower gate voltages, showing the splitting of the neutrality line into two curves. White lines, theoretically predicted neutrality curves ('Splitting of neutrality line' in Methods and Supplementary Fig. 8).

X-shaped splitting, consistent with the functional form predicted theoretically[20] (Fig. 2d inset, Supplementary Fig. 9 'Splitting of neutrality line' in Methods). To the right of each curve, the corresponding layer is electron-doped; and to the left, hole-doped, creating distinct regions in the map corresponding to pp, np, and nn layer-doping configurations. This reveals that hydrogenation takes place when at least one of the layers is electron doped. Time-resolved layer-selective electronic measurements under fixed gate voltages (Fig. 2c), showed that for regions in which only the top layer is strongly electron doped (box 1, Fig. 2c, d), the top layer was hydrogenated; and in regions in which both layers were strongly n-doped (box 2, Fig. 2c, d), both layers were hydrogenated. The map also reveals that the boundary of the hydrogenated region in the map runs alongside the neutrality curves of the top and bottom layers (marked with arrows in Fig. 2d).

These findings are readily understood from our analytical model of the charge density in the bilayer (Supplementary Fig. 8). The model shows that the boundaries correspond exactly to voltage combinations for which $n$ in one of the layers reaches $10^{14}\,cm^{-2}$. Hence, in the region containing box 1, only one layer has $n > 10^{14}\,cm^{-2}$, whereas in the region containing box 2, both layers do. This reveals that selective hydrogenation arises from electric-field-induced polarisation of the crystal, which drives $n$ in one layer above $10^{14}\,cm^{-2}$ and $\ll 10^{14}\,cm^{-2}$ in the

other. The highly charged layer then undergoes hydrogenation via the same mechanism demonstrated in our earlier work on monolayer graphene, where hydrogenation took place at this charge density[2]. These results thus explain our earlier observation that lower total carrier density requires higher electric fields—namely, under less total charge, stronger polarisation is necessary to accumulate the threshold charge for hydrogenation in one of the layers.

To demonstrate robust control over these processes, we used them to construct logic gates (Fig. 3). The two gate voltages were defined as logic inputs IN1 and IN2, and the resulting electronic and proton currents were monitored using three measurement configurations. In Mode I, we measured the in-plane electronic current from each graphene layer independently (logic outputs OUT1 and OUT2). When the top layer was hydrogenated and became insulating, its electronic current dropped to the off state; when dehydrogenated, it returned to the on state. This yielded a NOT logic gate ("NOT A") in the top layer with ON/OFF ratio of $10^4$, and symmetrically, a "NOT B" gate in the bottom layer. In Mode II, we measured the in-plane current of one layer (top or bottom) as OUT1 and the interlayer tunnelling current between the two layers as OUT2. A "NOT" logic gate was achieved for OUT1, just like in Mode I. However, now the tunnelling current provided a different logic gate. When either the top or

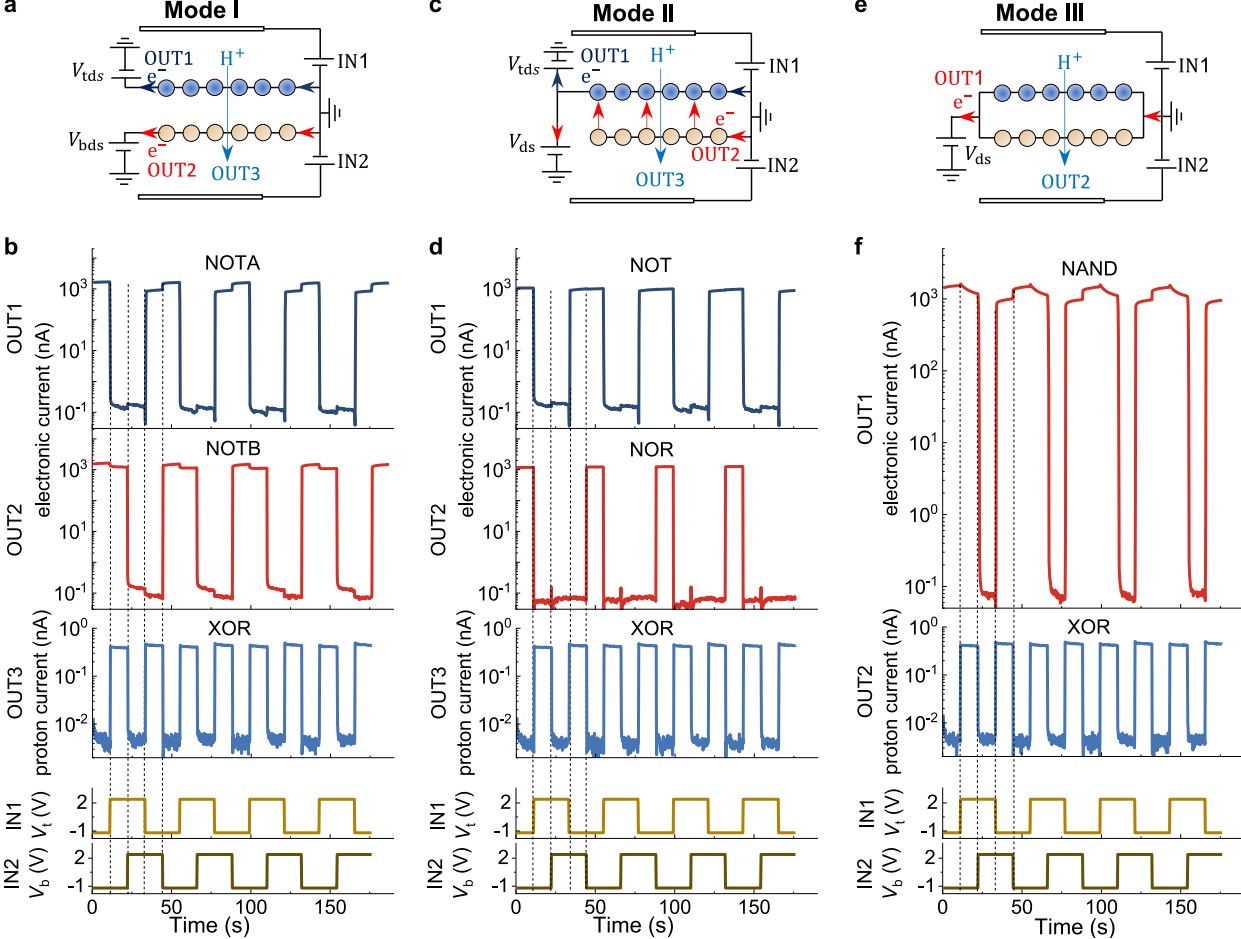

**Fig. 3 | Layer selective hydrogenation enables configurable parallel logic gates.** **a** Signal protocol for Mode I. **b** Independent in-plane electronic currents from top (OUT1, blue) and bottom (OUT2, red) graphene layers. Hydrogenation selectively turns off current in each layer, implementing parallel NOT A and NOT B in the top and bottom layers, respectively. Third row: proton current yields an XOR gate in parallel. Bottom two rows: input waveforms (IN1, IN2) cycle. **c** Signal protocol for Mode II. **d** In-plane electronic current from the top layer (OUT1, blue) and interlayer tunnelling current (OUT2, red). Response from top layer is the same as in (**a**).

Tunnelling is suppressed when either layer is hydrogenated, yielding a NOR universal gate. Third row, proton transport realises an XOR logic gate. **e** Signal protocol for Mode III. **f** Total electronic current (OUT1) is measured with both layers connected to the same source-drain voltage. Only hydrogenation of both layers switches the current off, realising a NAND universal gate. Second row, proton current realises an XOR gate. All modes were characterised with the same input waveforms. Dashed lines in all panels, guide to the eye.

bottom layer was hydrogenated, the tunnelling current dropped to the off state, functioning as a NOR gate (OUT2), which is a universal logic gate. In Mode III, the two layers were electrically connected, and we measured the total electronic current (OUT1). Only when both layers were hydrogenated did the current drop to a low state, thus implementing a NAND gate—another universal logic gate. In all three modes, the out-of-plane proton current provided an additional parallel XOR logic gate with ON/OFF ratio of $10^2$. These results therefore show that protons can be used to encode information in a configurable architecture that enables parallel logic operations within a single graphene device.

Our results introduce a new electrode-electrolyte interface with two decoupled 2D electron gases. We show that proton adsorption can be controlled by field-induced charge imbalances between individual layers, which is a new way to control chemical adsorption in surfaces and opens new pathways to drive electrochemical processes. We exploited the several degrees of freedom in this interface to enable configurable and parallel proton–electronic logic gates within a single device, an approach that could be extended to several of the 2D heterostructures being intensely researched in electron transport studies. Beyond this, the new interface demonstrated here could enable novel control protocols for other processes governed by electronic charge density, such as ion intercalation or redox processes, opening new design opportunities for energy technologies and ion-based information processing in van der Waals heterostructures.

## Methods

### Device fabrication
Apertures 10 μm in diameter were etched into silicon nitride substrates (comprising a 500 nm $SiN_x$ layer on both sides of a silicon wafer) using a combination of photolithography, wet etching, and reactive ion etching, as reported previously[9]. Two pairs of electrodes (Au/Cr) were patterned on the substrate using photolithography and electron beam evaporation. Two mechanically exfoliated graphene flakes, each with a rectangular shape (tens of micrometres in length and approximately 10 μm in width), were stacked on opposite sides of an hBN layer with a thickness of 10–20 nm. The hBN layer contained a circular hole, aligned in size and position with the hole in the underlying $SiN_x$ substrate (Supplementary Fig. 1). The graphene flakes were assembled in a cross configuration, centred on the circular hole, such that each flake was individually contacted by a pair of Au electrodes. In this design, the graphene bilayer is electrostatically gated from both sides, enabling double gating of the stacked structure. Crucially, the two graphene flakes are in physical contact with each other only in the central, gated region—defined by the aperture in the hBN and $SiN_x$ layers—where both flakes are exposed to the electrolyte. This configuration ensures that any changes in conductivity due to hydrogenation are detectable. Specifically, if one graphene layer becomes hydrogenated and thus insulating, electron transport across that layer is suppressed because the two flakes are not electrically connected outside the gated area. In contrast, without the hBN spacer, the flakes would be in contact over a larger area, allowing current to bypass the hydrogenated region through ungated or unexposed parts of the other flake, masking the effects of hydrogenation.

An SU-8 photo-curable epoxy washer with a 15 μm-diameter hole was transferred over the assembly and on the source and drain electrodes[12] with the hole in the washer aligned with the aperture in the silicon-nitride substrate (Supplementary Fig. 1) to ensure that the electrodes are electrically insulated from the electrolyte. The proton-conducting electrolyte used was 0.18 M bis(trifluoromethane)sulfonimide (HTFSI) dissolved in poly(ethylene glycol) (average molecular weight Mn of 600)[13]. Palladium hydride foils (-0.5 cm²) were used as gate electrodes. The device was placed inside a gas-tight chamber filled with argon for electrical measurements.

### Transport measurements
A dual-channel Keithley 2614B sourcemeter was used to independently bias the top and bottom gates, while a second dual-channel Keithley 2636 A sourcemeter was used to apply the drain–source bias and measure the in-plane electronic conductance of each graphene layer. We measured the electronic current of the devices in three different configurations (Modes I, II, and III). In Mode I, we recorded the conductance of each layer by applying drain-source voltages to each layer. In Mode II, we drove the interlayer tunnelling current, $I_{tun}$, by applying a voltage $V_{ds}$ between the top and bottom layer and then recorded the resulting tunnelling voltage ($V_{tun}$) with a Keithley 2182 A Nanovoltmeter. In Mode III, we measured the total in-plane electronic current by applying a single drain-source voltage to both layers. On the other hand, the proton current ($I$) was recorded by monitoring the gate currents on both the top and bottom channels, as previously reported[2]. All measurements were obtained using software that allowed controlling $V_t$ - $V_b$ and $V_t$ + $V_b$ as independent variables. To map the response of the samples, we swept $V_t$ - $V_b$ (for a fixed $V_t$ + $V_b$) at a rate of 10 mV s$^{-1}$ and stepped $V_t$ + $V_b$ with 10 mV intervals.

Note that previous works have established that in these devices the two gates are independent from each other[2,3]. We confirmed this here in the extended gate voltage range used in this work (Supplementary Fig. 3). This was done by adding a reference electrode to the top channel (a PdHx foil of the same size as the gate electrodes) and measuring its potential, $V_t^{ref}$, with a Keithley 2182 A Nanovoltmeter. These data showed that $V_t^{ref} = V_t$ for all values of $V_b$ within the experimental scatter of less than 5 mV. These data also demonstrate that the voltage drop takes place at the graphene electrode, with a negligible drop in the electrolyte, as expected for microelectrode devices[25].

### Raman spectroscopy
For Raman measurements, the samples were placed inside a gas-tight Linkam chamber (HFS600E-PB4), and spectra were collected using a 514 nm laser. Supplementary Fig. 2 shows that twisted bilayer graphene samples exhibit a blueshift in the 2D and G bands accompanied by a larger $I_{2D}/I_G$ ratio. A ratio of $I_{2D}/I_G < 2$ is associated with a twist angle within the range of 3°–15°[26–28], whereas $I_{2D}/I_G > 2$ indicates an angle within the range 15°–30°[26–28]. In both cases, the areal fraction of AA-stacked regions is estimated to be approximately constant at ≈30%[24]. Raman mapping across the entire sample area confirmed that the spectra were uniform across the sample, indicating a uniform twist angle throughout.

For in situ Raman measurements during the hydrogenation process, the spectra were acquired under constant gate voltages. The background signal from the electrolyte was subtracted for clarity, as reported previously[2,13]. Hydrogenation of a single graphene layer yields a strong D peak, similar to that observed for hydrogenated monolayer graphene previously[2,13] (Supplementary Fig. 5). On the other hand, while the 2D peak was less intense than in pristine twisted bilayers, it remained defined[2,13], as expected. Only if both layers were hydrogenated, the 2D peak was fully smeared. Crucially, the hydrogenation transition was fully reversible: the D peak disappeared completely from the Raman spectra following de-hydrogenation.

### Scanning transmission electron microscopy (STEM) imaging
For imaging, twisted bilayer graphene devices were fabricated on custom-made silicon nitride TEM grid coated with Ta (2 nm) & Au (20 nm) to enhance the adhesion of the stack to the grid and annealed at 400 °C under H₂/Ar with a pressure of 1.2 bar for 13 h. Atomic resolution high angle annular dark field (HAADF) STEM images were collected using an ThermoFisher Iliad Ultra STEM operated at 80 kV with an aberration-corrected probe, a 24 mrad probe convergence angle, and a HAADF inner angle of 80 mrad. STEM image analysis and filtering was performed using open-source python package Hyperspy version 2.3.0 [hyperspy v2.3.0 (Zenodo, 2025).].

## Estimation of $E$ and $n$

In our previous report[2], we detailed the derivation[29] of the well-established relations[4,5,29–31] between the top ($V_t$) and bottom ($V_b$) gate voltages and $E$ and $\mu$. For twisted bilayer graphene, we have the same relation for the geometrical capacitance term: $neC^{-1} = (V_t + V_b) - \Delta^{NP}$, where $\Delta^{NP} \equiv V_t^{NP} + V_b^{NP}$, $n = n_t + n_b$ is the total charge in the bilayer, $n_t$ ($n_b$) is the charge density in the top (bottom) layer, and $C$ is the area-normalised capacitance of the electrolyte (the value in our system is $\approx 20\ \mu F\ cm^{-2}$ at a scan speed of 10 mV/s)[2]. To consider the quantum capacitance of graphene, we note that $\mu_e = \hbar v_F (\sqrt{\pi|n_t|} + \sqrt{\pi|n_b|})$, with $v_F \approx 1 \times 10^6\ m\ s^{-1}$ the Fermi velocity in graphene, and $\hbar$ the reduced Planck constant. This yields the relation to[50]:

$$(V_t + V_b) - \Delta^{NP} = neC^{-1} + \hbar v_F e^{-1}(2\pi n)1/2 \quad (1)$$

The formula for $E$ is the same as in our previous work[2]:

$$E = E_t - E_b = C(2\varepsilon)^{-1}(V_t - V_b) \quad (2)$$

with $\varepsilon = \varepsilon_0\ \varepsilon_r$, $\varepsilon_0$ the vacuum permittivity and $\varepsilon_r \approx 10$ the solvent dielectric constant[32,33]. Introducing the numerical values of the parameters yields: $(V_t + V_b) - \Delta^{NP} \approx 0.8 \times 10^{-14}\ V\ cm^2\ n + 1.64 \times 10^{-7}\ V\ cm\ n^{1/2}$ and $E \approx 1.13 \times 10^9\ V\ m^{-1}\ (V_t - V_b)$.

## Splitting of the neutrality line

The graphene layers in bilayer graphene are modelled as two conducting thin plates which are electrostatically coupled by a capacitance $C_m$. This capacitance was determined in ref. [20] and found to be $C_m \approx 7.5\ \mu F\ cm^{-2}$. The electrolyte potential profile in the electrolyte is modelled by a Gouy-Chapman-Stern model[1] and results in an electrical double layer near the graphene surface. The gate potential almost exclusively drops over this Stern layer, which has a typical thickness[2] of about 0.9 nm. The electrolyte is modelled by a capacitor with $C_t \approx 20\ \mu F\ cm^{-2}$. Note that $C_t/C_m \sim 2.67$, and thus we are in a very different regime from the hBN encapsulated twisted bilayer graphene investigated in ref. [20] where $C_{hBN}/C_m \sim 0.11$. Supplementary Fig. 8 shows a schematic of the analytical model.

From Gauss' law, we obtain:

$$C_t(V_t - V_{Gt}) - C_m(V_{Gt} - V_{Gb}) = -en_t \quad (3a)$$

$$C_m(V_{Gt} - V_{Gb}) - C_b(V_{Gb} - V_b) = -en_b \quad (3b)$$

where $V_t$, $V_{Gt}$, $V_b$, $V_{Gb}$ are as defined in Supplementary Fig. 8b and the charge density on the graphene layers is determined by the Fermi energy $E_{Fi} = \pm\ \hbar v_F\ (\pi|n_i|)^{1/2}$ in the layers: $n_b = -(eV_{Gb})^2(\pi\hbar^2 v_F^2)^{-1}\text{sgn}(V_{Gb}) = -0.739 \times 10^{14}\ V_{Gb}^2\text{sgn}(V_{Gb})$ where $V_{Gb}$ is expressed in Volt and density in $cm^{-2}$. A similar expression is obtained for $n_t$. The neutrality line $n_t = 0$ is then obtained as:

$$V_b = -(\gamma + \beta)V_t - \alpha\beta\gamma\ V_t^2\ \text{sgn}(V_t).$$

With $\beta = C_t/C_b$ the asymmetry of the gates, $\gamma = C_t/C_m$ the ratio of electrolyte capacity versus the bilayer graphene capacitance, and $\alpha = e^3 C_m^{-1}\ (\hbar v_F \sqrt{\pi})^{-2} = C_m^{-1} \times 11.8\ \mu F\ cm^{-2}\ V^{-1}$. Similarly, we find for the other neutrality line $n_b = 0$ that:

$$V_b = \beta(2\alpha\gamma)^{-1}(1 + \gamma)\ \text{sgn}(V_t)\ \{1 - [1 + 4\alpha\gamma(1 + \gamma)^{-2}\ V_t\ \text{sgn}(V_t)]^{1/2}\}$$

These equations are plotted directly into the map in Fig. 2d using $C_t = C_b = 20\ \mu F\ cm^{-2}$ and $C_m = 7.5\ \mu F\ cm^{-2}$, which results in $\gamma = 2.67$, $\beta = 1$, and $\alpha = 1.57$.

## Analytical model

The voltage on the separate graphene layers ($V_{Gt}$, $V_{Gb}$) can be obtained from a self-consistent solution of the coupled non-linear equations Eq. (3a,b) which can be written as

$$\gamma(V_t - V_{Gt}) - (V_{Gt} - V_{Gb}) - \alpha V_{Gt}^2\text{sgn}(V_{Gt}) = 0 \quad (4a)$$

$$(V_{Gt} - V_{Gb}) - \gamma\beta^{-1}(V_{Gb} - V_b) - V_{Gb}^2\text{sgn}(V_{Gb}) = 0 \quad (4b)$$

From Eq. (4a), we solve for $V_{Gb}$ and insert it into Eq. (4b), which results in the following 4th order algebraic equation $ax^4 + bx^3 + cx^2 + dx + e = 0$ with $x = V_{Gt}$ and the coefficients are given by: $a = \alpha^3\ \text{sgn}(V_{Gb})$, $b = 2\alpha^2(\gamma + 1)\ \text{sgn}(V_{Gb})\ \text{sgn}(V_{Gt})$, $c = -2\alpha^2\gamma\ V_t\ \text{sgn}(V_{Gb})\ \text{sgn}(V_{Gt}) + \alpha(\gamma + 1)^2\ \text{sgn}(V_{Gb}) + \alpha(1 + \gamma\beta^{-1})\text{sgn}(V_{Gt})$, $d = -2\alpha\gamma(\gamma + 1)\ V_t\ \text{sgn}(V_{Gb}) + \gamma(1 + \gamma\beta^{-1} + \beta^{-1})]$, $e = \alpha\gamma^2\ V_t^2\ \text{sgn}(V_{Gb}) - \gamma(1 + \gamma\beta^{-1})V_t - \gamma\beta^{-1}\ V_b$. Inserting this solution into Eq. (4b) gives $V_{Gb}$.

Adding Eq. (3a) and Eq. (3b), we find for the total charge density on bilayer graphene under the assumption of $C_t = C_b = C$: $n = n_t + n_b = Ce^{-1}\{V_{Gt} + V_{Gb} - (V_t + V_b)\}$.

## DFT calculations

First-principles calculations were performed using VASP with plane-wave basis sets and the PAW method[34–37]. Van der Waals interactions were included using the DFT-D3 correction by Grimme[35], and spin polarization was considered. Exchange–correlation energies were treated within the GGA-PBE functional[36]. A 500-eV energy cutoff and Monkhorst-Pack k-point mesh with a resolution of $2\pi \times 0.05/\text{Å}$ were used. Convergence thresholds were $10^{-6}$ eV (energy) and 0.01 eV/Å (forces). The simulation cell contained a 6×6 graphene supercell (72 atoms/layer) with periodic boundary conditions and a vacuum spacing of 20 Å in the z-direction to avoid image interactions. Different stacking configurations (AA′, AB, SP) were tested, and all atoms were fully relaxed. Transition states for hydrogenation and proton permeation were identified using Nudged Elastic Band (NEB) calculations.

Supplementary Fig. 4a shows the energy profile for a proton permeating through the central ring of twisted bilayer graphene (AA stacking). The barrier to pass through the first layer is 1.4 eV, consistent with previous works[2,9]. After entering the interlayer space, the energy profile shows a local minimum at the centre. This increases the barrier to pass the second layer to 1.7 eV, further suppressing proton transport. Since the AA-stacked regions occupy only ~30% of the total area, the reduced areal coverage and the higher barrier account for the lower proton flux compared to monolayer graphene observed experimentally.

Supplementary Fig. 4b shows hydrogen adsorption barriers for AA and AB stacked bilayers. In twisted bilayer graphene, hydrogenation barriers range from 0.05–0.20 eV. AB stacking shows particularly low barriers (<0.1 eV), ~50% less than in monolayer graphene (~0.2 eV). In contrast, AA stacking exhibits barriers similar to monolayer graphene. Adsorption wells of 0.9–1.1 eV were found, lower than for monolayer graphene, due to interlayer electronic effects that reduce the stability of the chemisorbed state and facilitate dehydrogenation.

## Configurable parallel logic gate measurements

We implemented three logic modes using a single twisted bilayer graphene device by defining gate voltages $V_t$ and $V_b$ as logic inputs. Mode I uses in-plane currents from each graphene layer as outputs. Mode II uses both the in-plane electronic and interlayer tunnelling currents as output. Mode III uses total in-plane current. For all three modes, the proton current provides another parallel output. Logic operations were driven by square-wave gate voltages, with −1.2 and +2.4 V representing logic input 0 and 1. All three modes used the same input signals, enabling straightforward configuration of the different logic gates within the device. For the these modes, the logic gates

(NOT, NOR and NAND) using electron transport which based on the selective hydrogenation process show the ON/OFF ratios of $10^4$ with cycle-to-cycle variation less than 5%, while the logic gates (XOR) using proton transport show show the ON/OFF ratios of $10^4$ with cycle-to-cycle variation less than 10%.

To demonstrate the stability of the device during the selective hydrogenation and dehydrogenation process, a cyclability test were carried out based Mode I measurement configuration. For selective hydrogenation of the bottom layer while maintaining conductivity in the top layer, gate voltages of $V_t$ and $V_b$ were set to 0.9 V and 1.9 V, respectively. For dehydrogenation of the top layer, both $V_t$ and $V_b$ were set to be −1 V. Each hydrogenation–dehydrogenation cycle had a duration of 12.5 s.

## AB-stacked bilayers

AB bilayers display fundamentally different proton transport and adsorption phenomena. First, these crystals are impermeable to protons, even under large fields. We demonstrated as shown in Supplementary Fig. 7e and in our previous works[2,9]. Second, because the two graphene layers in AB stacking are electronically coupled, it is not possible to induce layer-selective conductor–insulator transitions. When a sufficiently large potential is applied to one side, the upper surface of the bilayer can be hydrogenated, which is reflected by a conductance drop at the corresponding potential (Supplementary Fig. 7c). However, the bilayer remains overall conductive. Only when a large potential is subsequently applied to the opposite side, does the system undergo a full insulating transition (Supplementary Fig. 7d). These control measurements show that twist-induced electronic decoupling is essential for enabling both layer-selective hydrogenation and logic functionality.

## Data availability

All relevant data are available from the corresponding authors and at https://zenodo.org/records/18783917.

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

## Acknowledgements

This work was supported by UKRI (EP/X017745: M.L.-H, EP/X041204: S.J.H., EP/S021531: S.S-S., EP/Y024303:G.T.), the Directed Research Projects Program of the Research and Innovation Center for Graphene and 2D Materials at Khalifa University (RIC2D-D001: M.L.-H., L.F.V. and

D.B.), The Royal Society (URF\R1\201515: M.L.-H.), the U.S. Army DEV-COM ARL Army Research Office (ARO) Energy Sciences Competency, (Electrochemistry or Advanced Energy Materials) Program award # W911NF-25-1-0041 (M.L-H.). The views and conclusions contained in this document are those of the authors and should not be interpreted as representing the official policies, either expressed or implied, of the U.S. Army or the U.S. Government. Part of this work was supported by the Flemish Science Foundation (FWO-Vl: F.M.P.) and FUNCAP and CNPq (312539/2025-8, 437067/2018-1, 423423/2021-5, 408144/2022-0: RNCF), and the European Research Council (Grant ERC-2016-STG-Evo-luTEM-715502, S.J.-H.). TEM access was supported by the Henry Royce Institute for Advanced Materials, funded through EPSRC grants EP/R00661X, EP/S019367, EP/P025021 and EP/P025498.

## Author contributions

M.L.-H. designed and directed the project with J.T. J.T., G.C., E.H., M.A., X.Z. & Y.M. fabricated devices. J.T. & H.L. performed the transport measurements and analysis with help from D.D and E.H. G.R.T., S.S.-A. & S.J.H. performed STEM characterisation and analysis. F.M.P. and D. R. C. performed analytical theory calculations. D.B. & L.F.V. performed DFT calculations. M.L.-H. & J.T. wrote the manuscript with input from all the authors.

## Competing interests

The authors declare no competing interests.

## Additional information

**Peer review information** : *Nature Communications* thanks Hyeon Han, Young-Jun Yu and the other, anonymous, reviewer(s) for their contribution to the peer review of this work. A peer review file is available

