## [Transparent Peer Review file · Nature Communications]

Layer-selective hydrogenation and proton transport in twisted bilayer graphene

Corresponding Author: Professor Marcelo Lozada-Hidalgo

Version 0:

Reviewer comments:

Reviewer #1

(Remarks to the Author)

This is a highly compelling study that introduces a novel electrode-electrolyte interface based on twisted bilayer graphene. It elegantly demonstrates how the decoupling of electronic systems in a van der Waals heterostructure can be harnessed for precise electrochemical control and configurable logic operations. Several issues regarding device validation, performance quantification, and operational stability will be essential for establishing its full impact and technological relevance. Therefore, a minor revision is necessary.

1. The device configuration described in Extended Data Figure 1 shows that the two graphene layers are separated by a h-BN spacer and only make contact within a limited suspended area. This raises questions about whether the system truly functions as a coherent twisted bilayer graphene system or rather as two independent monolayers with partial contact. Additional evidence, such as direct observation of Moiré patterns in tunneling spectroscopy, would help validate the twisted bilayer characterization and clarify how the electronic coupling in the contact region dominates the overall system behavior.
2. The 24-hour state retention under constant gate voltages (Figure 1f) is impressive for memory applications, but the practical implementation requires demonstration of cyclical stability. While Raman data in Extended Data Fig. 5b suggests some reversibility, complementary electronic transport data showing multiple hydrogenation-dehydrogenation cycles are lacking. Quantification of performance metrics such as ON/OFF ratio degradation over cycles would significantly strengthen the application potential claims.
3. The logic gate demonstration in Figure 3 is conceptually interesting, but lacks quantitative performance benchmarks. Explicit documentation of ON/OFF current ratios for each output under standardized conditions would enable proper evaluation of the logic operation quality. Additionally, characterization of switching speed and cycle-to-cycle variation would provide crucial information for assessing practical applicability.
4. While the manuscript highlights the advance beyond previous work on monolayer graphene, the introduction could more precisely articulate the specific conceptual leap achieved by the decoupled bilayer architecture. The outlook section would benefit from concrete examples of other 2D heterostructures where this approach could be applied, such as twisted transition metal dichalcogenide systems for ion intercalation control.

Reviewer #2

(Remarks to the Author)

The authors report layer-selective protonation in twisted bilayer graphene using double electrolyte gating and further demonstrate configurable logic operations. Independent control of the electric field and carrier density is claimed to modulate the resistance of each layer. The results are intriguing, but several aspects of the mechanism and data interpretation require clarification, as follows:

1. The authors analyze their data based on an electric field defined as $E \propto V_t - V_b$. However, this definition is not physically reasonable in the double electrolyte gating. I suggest analyzing the results in terms of the individual gate voltages (V_t and V_b), both their sign and magnitude, for the following reasons. The electrolyte contains both cations and anions. Under positive bias, H^+ ions migrate toward the graphene, while under negative bias, $TFSI^-$ or O_2^- ions move toward the interface. $TFSI^-$ ions induce mainly electrostatic effects, whereas O_2^- ions can lead to both electrostatic and electrochemical modifications of graphene. Thus, both types of negatively charged species affect the conductivity of graphene.

When analyzing Fig. 2 by decomposing the data into V_t and V_b components, both top and bottom graphene layers show abrupt conductance decreases at approximately V_t (or V_b) = ~ 2 V. This appears to be a critical voltage corresponding to H^+ migration into the respective layer. Therefore, the electric potential and field should be defined separately for each layer, rather than using a global $V_t - V_b$ term, which is not sufficiently meaningful to describe layer-specific conductance changes.

2. The assumption that carrier density is simply proportional to $V_t + V_b$ is also not reasonable. This model implicitly assumes that only H^+ ions participate in gating. However, when V_t or V_b is negative, migration of TFSI⁻ or O_2^- ions should also be considered. It would therefore be more reasonable to define and evaluate carrier density individually for each layer.

3. The carrier density is estimated by simple equations, but such estimates are often inaccurate for electrolyte gating. Ionic gating typically involves both electrostatic accumulation and electrochemical reactions, and electric-double layers are formed on the surface. This means that carrier density cannot be defined solely by applied voltage. I recommend directly measuring carrier density using Hall measurements or an equivalent experimental approach.

4. The authors attribute independent modulation of the top and bottom layer resistances to electronic decoupling induced by the large twist angle. However, the observed independence could simply result from the independent control of V_t and V_b for each layer, rather than from the twisted structure itself. It remains unclear whether the twist angle plays any essential role. I suggest presenting twist-angle-dependent measurements to support this claim.

5. TFSI-based ionic liquids are known to contain residual water, which allows migration of both H^+ and O_2^- ions. Please refer to previous reports showing H^+ migration under positive bias and O_2^- migration under negative bias (Nature 2017, 546, 124–128; ACS Nano 2022, 16, 6206–6214). H-TFSI systems are also hygroscopic, enabling H^+ and O_2^- migration during gating. Therefore, it is highly possible that O_2^- ions migrate into graphene under negative V_t or V_b .

6. The manuscript states that the top and bottom graphene layers are in contact within the aperture of the hBN layer. Please clarify the thickness of the hBN spacer and whether the two graphene layers are indeed in electrical contact. The data in Fig. 2 suggest connection between layers, but the term “tunneling current” used to describe the measurement is confusing. It seems more appropriate to describe this simply as out-of-plane current between the two graphene sheets rather than tunneling through a barrier.

Reviewer #3

(Remarks to the Author)

The authors demonstrate hydrogenation and proton transport in twisted bilayer graphene. Based on AA stacked region in twisted bilayer by employed hBN spacer with hole, the conductance variation of bilayer as a function of electric field of hydrogenation/de-hydrogenation and top, bottom gate voltage (V_t and V_b). Since this manuscript introduces a layer selective proton transport characterization as well as its logic application of twisted bilayer graphene, this result will be a good reference on electrolyte gating graphene research field. However, because the mechanically stacked bilayer graphene leads to various complex surface condition comparing with their previous paper about electrolyte gating study for monolayer graphene (Ref. 2), more detail experimental results and explanations should be supplemented for publication. The detailed comments are as follows.

1. During mechanically stacking of graphene layers, undesired bubbles and wrinkles often arise and can influence conductance variability and proton transport in stacked bilayer graphene. Because authors analyze AA and AB stacking in twisted bilayer graphene (TBG) including the associated area fractions and specific twist angles, it is important to distinguish intrinsic stacking effects from extrinsic disorder. To address this, please provide additional evidence and a deeper explanation demonstrating that the contact/interfacial area of the twisted bilayer is clean and representative. In particular, direct structural characterization (e.g., HRTEM) to map AA/AB domains without undesired bubbles and wrinkles would substantially strengthen the manuscript.

2. Authors propose that AA-stacked bilayer graphene hydrogenates at a lower charge-density threshold than monolayer graphene (Ref. 2) and attribute this to electric-field differences. For clarity, please articulate the discussion of which physical factors most plausibly account for the reduced threshold. Framing the observation within these mechanisms would strengthen the manuscript.

3. For finding neutrality line, authors employed calculation with capacitance values extracted from references. Furthermore, since the extended data Fig. 6 exhibits CNP for natural (Bernal) stacked bilayer, we can easily observe the single CNP for Bernal (AB) stacked bilayer graphene. However, if there are layer selected conductance variation for top and bottom graphene, we could observe charge neutrality position (CNP) of transport curve for top and bottom monolayer graphene as a function of V_t or V_b , respectively. Furthermore, we will observe more mixed CNP behavior dependent on twisted bilayer graphene area. I can also see CNP shapes as a function of E for conducting condition of top and bottom layer in Fig. 1 c and d. If authors exhibit externally experimental CNP behavior as well as discussion about it, their manuscript will be more strengthened, too.

4. There should be mentions for scale bar sizes in extended data Fig.2 b, c and Fig. 6 a.

Reviewer #4

(Remarks to the Author)

The manuscript presents an elegant and conceptually innovative study of electrochemical hydrogenation and proton transport in twisted bilayer graphene (TBG) under dual ionic gating. Building on the authors' recent paper in Nature demonstrating decoupled control of proton transport and hydrogenation in monolayer graphene, this work extends the paradigm to twisted bilayers, exploiting twist-induced electronic decoupling to achieve layer-selective hydrogenation and logic operations within a single device. This represents a significant advance: by leveraging the independent carrier densities emerging from the two-layer electronic systems of large-angle TBG, the authors introduce a new type of electrode–electrolyte interface in which electrochemical processes can be spatially and electronically partitioned at room temperature. The demonstration of configurable NOT, NOR, NAND, and XOR logic elements—arising from a single material platform and controlled solely via gate voltages—is very nice, and opens promising avenues for ion–electron hybrid information processing.

The experimental methodology is sound, and the characterization (conductance mapping, tunneling measurements, Raman spectroscopy, and DFT calculations) provides a largely coherent picture of the underlying processes. The work is also clearly novel relative to the group's earlier monolayer study: the ability to target hydrogenation to an individual graphene layer at fixed overall charge density, and to use this selectivity as a functional device degree of freedom, is new, nontrivial, and likely to be of broad interest to the community.

That said, two aspects of the study would benefit from further clarification and tempering of claims.

1. Lack of systematic controls on untwisted (AB-stacked) bilayer graphene.

The manuscript includes only a minimal control measurement on AB-stacked bilayers—namely, a conductance map showing a single neutrality line. While this does verify the expected absence of layer decoupling, it does not test whether hydrogenation, proton transport, or logic functionality differ qualitatively from the twisted case. Because the main claims rely on the emergence of layer-selective behavior due to twist-induced electronic isolation, a more complete comparison with untwisted bilayers would significantly strengthen the conclusions. Even basic measurements—such as whether AB bilayers undergo single-layer or simultaneous dual-layer hydrogenation under equivalent fields, or whether proton transport is observable at all—would help establish that the observed phenomena are uniquely rooted in twist-angle physics rather than more general features of bilayer graphene under high electric fields.

2. Interpretation of the AA-stacked regions as the exclusive proton-transport pathway.

The manuscript repeatedly attributes proton permeation primarily to AA regions of the Moiré superlattice. Although this interpretation is physically plausible and supported by DFT barrier calculations, it remains hypothetical within the context of the experimental data presented. The proton-current measurements are spatially unresolved, and no experiment directly evidences transport localization to AA regions or correlates permeation with twist-angle-dependent AA area fraction. I recommend that the authors moderate the certainty of these claims and clarify that the AA contribution, while consistent with theoretical expectations, is not experimentally demonstrated. Future work—such as twist-angle dependence studies, spatially patterned gating, or correlated imaging—could substantiate this point.

In summary, this is a strong manuscript that presents noteworthy advances in electrochemical control of two-dimensional heterostructures. With a modest softening of the AA-region interpretation and consideration of additional bilayer-control measurements, the paper will provide a clearer and more rigorously supported account of an important new electrochemical phenomenon in twisted bilayer graphene. I find the work to be of high quality and recommend publication after these points are addressed.

Version 1:

Reviewer comments:

Reviewer #1

(Remarks to the Author)

The authors have provided a comprehensive and persuasive response to the initial review, significantly strengthening the manuscript through additional experiments, characterization, and discussion. This study remains a highly compelling demonstration of a novel electrode–electrolyte interface concept using twisted bilayer graphene. All major concerns raised have been effectively addressed:

1. The crucial question regarding the coherent bilayer nature of the interface has been conclusively resolved with the addition of atomic-resolution STEM imaging, which directly reveals the moiré pattern. This evidence, combined with the existing transport, Raman, and proton-blocking data, solidly validates the twisted bilayer system.
2. The practical applicability of the memory function is now strongly supported by the new cyclability data showing >1000 reproducible hydrogenation-dehydrogenation cycles with stable performance, moving beyond the initial impressive state retention.
3. The logic gate demonstrations are now underpinned by quantitative performance benchmarks, including ON/OFF ratios and cycle variability under standardized conditions. The discussion of switching speed, supported by prior work, adds crucial context for assessing potential applications.
4. The manuscript now more precisely articulates the conceptual leap enabled by the decoupled bilayer architecture and provides a concrete outlook on extending this approach to other 2D heterostructures.

Reviewer #2

(Remarks to the Author)

The authors have revised the manuscript and addressed most of the reviewer comments. However, the key claims of the manuscript are still not supported by sufficient experimental evidence, and therefore I cannot recommend publication of the current version.

The thickness of the hBN spacer is only 20 nm, while the width of the hBN aperture appears to be on the order of $\sim 10 \mu\text{m}$. Under this geometry, it is highly plausible that the top and bottom graphene layers come into physical contact at the center of the aperture. If such physical contact occurs, the electron transport between the top and bottom graphene layers should not be described as tunneling current. Therefore, the authors must provide direct experimental evidence demonstrating that the top and bottom graphene layers are not in physical contact within the aperture region. Otherwise, the use of the term "tunneling current" is not justified and should be avoided throughout the manuscript.

Reviewer #3

(Remarks to the Author)

The authors have adequately addressed the reviewers' concerns. Therefore, I recommend this manuscript for publication in Nature Communications.

Reviewer #4

(Remarks to the Author)

The response to referees is substantive, and I think the manuscript is acceptable for publication.

Version 2:

Reviewer comments:

Reviewer #1

(Remarks to the Author)

After carefully examining the persistent technical issues, I agree with the authors' adoption of quantum tunneling to explain the electron transport across the vdW gaps between two fully electronically decoupled graphene. Although lack of direct evidence in this study, the concept is in line with recent understanding of decoupled graphene with sufficient spatial separation of the Dirac cones. Further verification of the transport nature, although interesting, is clearly beyond the scope of this work.

Overall, I believe that the author has established a novel maneuverability of hydrogenation and proton transport in bilayer graphene system, and more importantly its practical implications in function devices. This work can be published at its current form.

In addition, I do suggest a clearer presentation of the key concept, device structure and fabrication, as the paper itself may be challenging to read for broader audiences.

Response to comment from Reviewer #1

This is a highly compelling study that introduces a novel electrode-electrolyte interface based on twisted bilayer graphene. It elegantly demonstrates how the decoupling of electronic systems in a van der Waals heterostructure can be harnessed for precise electrochemical control and configurable logic operations. Several issues regarding device validation, performance quantification, and operational stability will be essential for establishing its full impact and technological relevance. Therefore, a minor revision is necessary.

We are very grateful to the Reviewer for this encouraging assessment of our work. We have performed additional measurements, device characterisation and included new discussion to address the points raised.

1. The device configuration described in Supplementary Figure 1 shows that the two graphene layers are separated by a h-BN spacer and only make contact within a limited suspended area. This raises questions about whether the system truly functions as a coherent twisted bilayer graphene system or rather as two independent monolayers with partial contact. Additional evidence, such as direct observation of Moiré patterns in tunneling spectroscopy, would help validate the twisted bilayer characterization and clarify how the electronic coupling in the contact region dominates the overall system behavior.

The two graphene layers make direct contact across the entire suspended region, whose area is almost identical to their full geometric overlap (Supplementary Fig. 1c). Three independent measurements confirm that the system operates as a coherent twisted bilayer rather than two decoupled monolayers. First, the transport maps show a non-trivial splitting of the neutrality lines with an interlayer capacitance of $7.5 \mu\text{F cm}^{-2}$, consistent with theoretical expectations for misaligned twisted bilayer graphene and with previous experimental results (ref. 20). Second, the Raman spectra of the suspended region match those reported for twisted bilayer graphene (Supplementary Fig. 2, ref. 26-28). Note that the spectra do not display hBN bands, which would be apparent if hBN was present. Third, proton transport through the bilayer requires direct graphene-graphene contact and would be impossible if the layers were separated by as little as 4 layers of hBN, which are known to block proton flow (ref. 9).

Nevertheless, following the Reviewer comment we have now performed scanning transmission electron microscopy (STEM) analysis of our twisted bilayer graphene samples to confirm contact of the individual graphene layers. Atomic resolution imaging of the sample reveals a clear moiré pattern resulting from the direct contact of the individual graphene layers within the bilayer with a measurable interlayer twist angle of 12.5° and a corresponding moiré period of $1.18 \pm 0.06 \text{ nm}$. The atomic resolution HAADF STEM image has been incorporated in Fig. 1a in the revised manuscript (see also point #1 from Reviewer 3).

2. The 24-hour state retention under constant gate voltages (Figure 1f) is impressive for memory applications, but the practical implementation requires demonstration of cyclical stability. While Raman data in Supplementary Fig. 5b suggests some reversibility, complementary electronic transport data showing multiple hydrogenation-dehydrogenation cycles are lacking. Quantification of performance metrics such as ON/OFF ratio degradation over cycles would significantly strengthen the application potential claims.

Following the reviewer comment, we have performed a cyclability test (Supplementary Fig. 6 in revised manuscript) and discussed the performance metrics in main text and the section “Configurable parallel logic gate measurements”. We found that the selective hydrogenation process is highly reproducible, with >1000 cycles with stable ON/OFF ratio of >10⁴ and less than 6% cycle-to-cycle variation. Such performance is within order of magnitude of promising electronic floating-gate 2D logic-in-memory devices, which achieve >5,000 ON/OFF cycles [Nature, 2020, 587(7832): 72-77].

3. The logic gate demonstration in Figure 3 is conceptually interesting, but lacks quantitative performance benchmarks. Explicit documentation of ON/OFF current ratios for each output under standardized conditions would enable proper evaluation of the logic operation quality. Additionally, characterization of switching speed and cycle-to-cycle variation would provide crucial information for assessing practical applicability.

Following the Reviewer’s comment, we have characterised the ON/OFF current ratios and cycle variability under standardised conditions (see also point #2) and now report these results in the Methods section “Configurable parallel logic gate measurements”. In addition, we have performed dedicated measurements of the hydrogenation switching speed, which we reported in a manuscript a couple of months ago (ref. 23 in revised manuscript). Fig. R1 (adapted from our recent manuscripts) shows that the switching time depends strongly on

the applied potential, ranging from seconds at 1.7 V vs NP to less than one microsecond. The limiting switching speed is around 10 times the RC time constant of the electrochemical circuit, which arises from the formation of the electrochemical double layer.

Fig. R1 | Device switching speed based on hydrogenation process. Time necessary to achieve the insulating transition as a function of V_G (left y-axis). Right y-axis, corresponding proton adsorption rate. Triangle marks the rate of acceleration of the switching speed with gate voltage. Vertical dashed line marks the minimum potential required to observe

the hydrogenation transition.

4. While the manuscript highlights the advance beyond previous work on monolayer graphene, the introduction could more precisely articulate the specific conceptual leap achieved by the decoupled bilayer architecture. The outlook section would benefit from concrete examples of other 2D heterostructures where this approach could be applied, such as twisted transition metal dichalcogenide systems for ion intercalation control.

Following the Reviewer comment we have now articulated the key conceptual leap achieved by our devices and the potential of similar structures for other chemical processes, such as intercalation in the conclusions/outlook page 6, last paragraph.

In brief, in a classical electrode–electrolyte interfaces, interfacial processes arise from controlling a single electronic system. Here, we introduce a new class of interface formed by two electronically independent layers, which enables access to regimes that are unavailable otherwise. Specifically, at fixed carrier density n but varying electric field E , a sufficiently strong E polarises the bilayer and redistributes charge between the layers, driving conductor–insulator transitions. This is in stark contrast to monolayers, where such transitions are controlled solely by n . The presence of these additional control variables introduces new reaction pathways for interfacial chemistry and opens opportunities for proton- and ion-based logic and memory architectures. Similar concepts could be extended to other layered systems, such as TMDC bilayers, where twisting or inserting thin hBN spacers may lead to rich intercalation phenomena by controlling independent electronic systems in a 2D heterostructure.

Response to comment from Reviewer #2

The authors report layer-selective protonation in twisted bilayer graphene using double electrolyte gating and further demonstrate configurable logic operations. Independent control of the electric field and carrier density is claimed to modulate the resistance of each layer. The results are intriguing, but several aspects of the mechanism and data interpretation require clarification, as follows:

We thank the Reviewer for taking the time to review our manuscript. We have done our best to address the points raised.

1. The authors analyze their data based on an electric field defined as $E \propto V_t - V_b$. However, this definition is not physically reasonable in the double electrolyte gating. I suggest analyzing the results in terms of the individual gate voltages (V_t and V_b), both their sign and magnitude, for the following reasons.

The electrolyte contains both cations and anions. Under positive bias, H^+ ions migrate toward the graphene, while under negative bias, $TFSI^-$ or O_2^- ions move toward the interface. $TFSI^-$ ions induce mainly electrostatic effects, whereas O_2^- ions can lead to both electrostatic and electrochemical modifications of graphene. Thus, both types of negatively charged species affect the conductivity of graphene.

When analyzing Fig. 2 by decomposing the data into V_t and V_b components, both top and bottom graphene layers show abrupt conductance decreases at approximately V_t (or V_b) = ~ 2 V. This appears to be a critical voltage corresponding to H^+ migration into the respective layer. Therefore, the electric potential and field should be defined separately for each layer, rather than using a global $V_t - V_b$ term, which is not sufficiently meaningful to describe layer-specific conductance changes.

2. The assumption that carrier density is simply proportional to $V_t + V_b$ is also not reasonable. This model implicitly assumes that only H^+ ions participate in gating. However, when V_t or V_b is negative, migration of $TFSI^-$ or O_2^- ions should also be considered. It would therefore be more reasonable to define and evaluate carrier density individually for each layer.

The Reviewer raises two points. First, that describing our devices using $V_t + V_b$ and $V_t - V_b$ is not physically meaningful. Second, that the role of TFSI⁻ and O₂⁻ is not considered. We respectfully disagree based on our results, theory predictions, and experimental work from independent groups, as we outline below.

Fig. R2.1 Fixed Conductance of monolayer graphene at constant $V_t + V_b$. Transfer curve of in-plane conductance as a function of $V_t - V_b$ for $V_t + V_b$ of -2.3, -1.8, 0 and 1 V.

We begin with the first point. In double-gated 2D crystals, the carrier density is set by $V_t + V_b$. This is well established for devices employing both crystalline and electrolyte gates and has been explored extensively in refs. 2–8. Importantly, this

is not a modelling assumption in our work, but an experimentally measured result in our devices. This is most clearly demonstrated in monolayer graphene. Fig. R2.1 shows that the electronic conductivity remains fixed for a given $V_t + V_b$ over the full range of $V_t - V_b$. The same principle holds for bilayer devices. In this case, $V_t + V_b$ fixes the total charge in the system, while $V_t - V_b$ controls how this fixed charge is distributed between the two layers. This is directly evidenced by the splitting of the neutrality lines, in quantitative agreement with theoretical predictions and prior transport experiments (ref. 20). While hydrogenation can indeed be induced by applying a sufficiently positive potential, this description is incomplete in a double-gated geometry. For example, setting one gate to +1.2 V induces hydrogenation in monolayer graphene when the second gate is held at 0.5 V (red dot, Fig. R2.2), but not when it is set to -1.4 V (blue dot, Fig. R2.2). This behaviour cannot be explained by a potential picture but is fully consistent with a process governed by the carrier density n , which is controlled by $V_t + V_b$.

Fig. R2.2 Maps of conductance show the coupling effect from top and bottom gates. Map of in-plane conductance of monolayer graphene as a function V_t and V_b . The white dashed line marks the V_t of 1.2 V. The red and blue dots mark the point when V_b is set to 0.5 V and -1.4 V, respectively, at $V_t = 1.2$ V.

The second point, concerning the possible role of other electrolyte ions (cations or anions), is addressed in point #5 below. In brief, reference experiments in which HTFSI is replaced by LiTFSI—containing all the ions mentioned by the Reviewer except protons—show none of the effects reported here.

3. The carrier density is estimated by simple equations, but such estimates are often inaccurate for electrolyte gating. Ionic gating typically involves both electrostatic accumulation

and electrochemical reactions, and electric-double layers are formed on the surface. This means that carrier density cannot be defined solely by applied voltage. I recommend directly measuring carrier density using Hall measurements or an equivalent experimental approach. As discussed above, the fact that the charge density is set by $V_t + V_b$ is an experimental observation. We then modelled this behaviour using analytical formulas based on the measured conductivity, which make no assumptions on the ions involved, only on the Debye length of the electrolyte and the solvent dielectric constant. While Hall effect measurements would more precisely determine the carrier density, such precision is not required to support the conclusions of this work. We therefore view these measurements as valuable future follow-up rather than essential to the present study.

4. The authors attribute independent modulation of the top and bottom layer resistances to electronic decoupling induced by the large twist angle. However, the observed independence could simply result from the independent control of V_t and V_b for each layer, rather than from the twisted structure itself. It remains unclear whether the twist angle plays any essential role. I suggest presenting twist-angle-dependent measurements to support this claim.

The decoupling induced by the large twist angle is essential to observing independent modulation of the top and bottom layer resistances. This is evident when compared with naturally AB-stacked bilayer graphene, where the layers are strongly hybridised and their conductivities cannot be measured separately. In the AB-stacked case the system behaves as a single electronic unit, and measuring each layer's conductivity is impossible. While it would be interesting to study the twist angle dependence of the process, such extensive study would overload our already extensive manuscript. We therefore view these measurements as interesting future follow-up work.

5. TFSI-based ionic liquids are known to contain residual water, which allows migration of both H^+ and O_2^- ions. Please refer to previous reports showing H^+ migration under positive bias and O_2^- migration under negative bias (Nature 2017, 546, 124–128; ACS Nano 2022, 16, 6206–6214). H-TFSI systems are also hygroscopic, enabling H^+ and O_2^- migration during gating. Therefore, it is highly possible that O_2^- ions migrate into graphene under negative V_t or V_b .

Fig. R2.3 | Devices measured using LiTFSI or measured at high negative gate voltage do not display a D band. a, Raman spectra collected from devices measured with LiTFSI (red) and HTFSI (blue) electrolyte. The Raman spectrum for LiTFSI electrolyte does not display a D band at high V bias. **b,** Raman spectra collected from devices measured with

HTFSI electrolyte at high negative V bias, ruled out the effects from TFSI or O_2^- species. The two results demonstrating that this D band arises exclusively due to proton adsorption in graphene.

The possibility that other ions besides protons are behind the conductor-insulator transition in graphene is ruled out by control experiments in which we used the same solvent and preparation methods but exchanged HTFSI for LiTFSI. No conductor-insulator transition nor D band is observed in the Raman spectra of the samples for positive potentials (Fig. R2.3a). These data were reported in Supplementary Fig. 4 in ref. 2, and we include it here as Fig. R2.3a. Moreover, Fig. R2.3b, shows that when negative gate voltage of -1.2 V is applied with HTFSI, no D peak appears, ruling out TFSI⁻ or O₂⁻ species as the source of our observations.

6. The manuscript states that the top and bottom graphene layers are in contact within the aperture of the hBN layer. Please clarify the thickness of the hBN spacer and whether the two graphene layers are indeed in electrical contact. The data in Fig. 2 suggest connection between layers, but the term “tunneling current” used to describe the measurement is confusing. It seems more appropriate to describe this simply as out-of-plane current between the two graphene sheets rather than tunneling through a barrier.

The hBN spacer away from the gated region is 20 nm thick and within the contact region there is no hBN at all. The two graphene layers are in direct electrical contact in the suspended region. Regarding the use of term ‘tunnelling current’, we respectfully point out that because electrons transfer through the vacuum space between the layers, the transport is due to tunnelling. For this reason, we have retained this term in the manuscript but have clarified that the layers are indeed in direct contact.

Response to comment from Reviewer #3

The authors demonstrate hydrogenation and proton transport in twisted bilayer graphene. Based on AA stacked region in twisted bilayer by employed hBN spacer with hole, the conductance variation of bilayer as a function of electric field of hydrogenation/dehydrogenation and top, bottom gate voltage (V_t and V_b). Since this manuscript introduces a layer selective proton transport characterization as well as its logic application of twisted bilayer graphene, this result will be a good reference on electrolyte gating graphene research field. However, because the mechanically stacked bilayer graphene leads to various complex surface condition comparing with their previous paper about electrolyte gating study for monolayer graphene (Ref. 2), more detail experimental results and explanations should be supplemented for publication. The detailed comments are as follows.

We thank the Reviewer for taking the time to review our manuscript. Below, we present additional characterisation and discussions to address the points realised.

1. During mechanically stacking of graphene layers, undesired bubbles and wrinkles often arise and can influence conductance variability and proton transport in stacked bilayer graphene. Because authors analyze AA and AB stacking in twisted bilayer graphene (TBG) including the associated area fractions and specific twist angles, it is important to distinguish intrinsic stacking effects from extrinsic disorder. To address this, please provide additional evidence and a deeper explanation demonstrating that the contact/interfacial area of the twisted bilayer is clean and representative. In particular, direct structural characterization

(e.g., HRTEM) to map AA/AB domains without undesired bubbles and wrinkles would substantially strengthen the manuscript.

Following the Reviewer's recommendation, we conducted atomic resolution high angle annular dark field (HAADF) STEM imaging of the devices in collaboration with our colleague Prof. Sarah Haigh. The images show that the devices are free from significant contamination and display the expected hexagonal Moiré pattern with the hexagonal distribution of AA and AB stacking regions (Fig. R3.1a&b). The new images are included in Fig. 1a in the revised manuscript.

Fig. R3.1 | TEM images of the twist bilayer graphene. **a**, Atomic resolution high angle annular dark field scanning transmission electron microscopy image of a representative bilayer graphene stack with a twist angle (θ) of 12.5° directly show the AA/AB domains. Scale bar, 2 nm. **b**, The corresponding FFT image shows the offset by the interlayer twist angle. Scale bar, 5 nm^{-1} .

2. Authors propose that AA-stacked bilayer graphene hydrogenates at a lower charge-density threshold than monolayer graphene (Ref. 2) and attribute this to electric-field differences. For clarity, please articulate the discussion of which physical factors most plausibly account for the reduced threshold. Framing the observation within these mechanisms would strengthen the manuscript.

This is an important point and goes to the core of our main finding. In the experiments, we fixed the total electron charge density in the bilayer at a level too low to trigger hydrogenation of the crystal. By increasing the electric field, the system becomes polarised: electrons and holes are driven towards opposite graphene layers. As a result, one layer accumulates electrons while the other becomes increasingly hole doped. When the electric field is sufficiently strong, the electron density in one layer reaches the threshold for hydrogenation, while the opposite layer is pushed further into p-type doping. Importantly, the total charge in the bilayer remains constant; only its distribution between the two layers changes as a result of the polarising electric field.

Fig. R3.2 | Electrostatic model of twisted bilayer graphene. **c**, Charge density in top layer as a function of gate voltages. Black curves, contour curves. Blue curve, $n_t = 1 \times 10^{14} \text{ cm}^{-2}$ contour curve. **d**, Corresponding charge density map for bottom layer. **e**, Total charge density in the

bilayer. Blue curve, $n_{total} = 2 \times 10^{14} \text{ cm}^{-2}$. For the above, $V_t + V_b \approx 1 \text{ V}$ marked by dashed line as discussed.

This behaviour is captured by our analytical model for the bilayer charge density (Supplementary Fig. 8), shown here as Fig. R3.2 for clarity. The left and middle panels show the charge density in the top and bottom layers, respectively, while the right panel shows the total charge density. Blue lines mark the threshold for hydrogenation. For $V_t + V_b \approx 1 \text{ V}$ (dashed line), hydrogenation does not occur at $V_t - V_b = 0$. Increasing $V_t - V_b$ along this line polarises the bilayer, increasing the charge density in the top layer until the hydrogenation threshold is crossed at $V_t - V_b \approx 4$. The bottom layer is simultaneously p-doped and therefore does not hydrogenate. Importantly, the total charge density remains constant along this trajectory (panel e).

3. For finding neutrality line, authors employed calculation with capacitance values extracted from references. Furthermore, since the Supplementary Fig. 6 exhibits CNP for natural (Bernal) stacked bilayer, we can easily observe the single CNP for Bernal (AB) stacked bilayer graphene. However, if there are layer selected conductance variation for top and bottom graphene, we could observe charge neutrality position (CNP) of transport curve for top and bottom monolayer graphene as a function of V_t or V_b , respectively. Furthermore, we will observe more mixed CNP behavior dependent on twisted bilayer graphene area. I can also see CNP shapes as a function of E for conducting condition of top and bottom layer in Fig. 1 c and d. If authors exhibit externally experimental CNP behavior as well as discussion about it, their manuscript will be more strengthened, too.

We observed the splitting of the neutrality lines experimentally for our twisted bilayer graphene samples. This is shown in Fig. 2d inset in the main text, where we fitted these data with the theory lines. Fig. R3.3 shows the experimental data without the theory lines, as well as cuts from the map, to highlight that the neutrality lines are directly visible. The splitting of the neutrality line is fully consistent with the discussion in point #2. Namely, in the double gated configuration, $V_t + V_b$ fixes the total charge density of the bilayer, whereas the electric field polarises the bilayer and thus redistributes charge between the layers. This phenomenon is fully accounted for by our analytical theory, which correctly predicts the position of the neutrality lines. These data has been added as Supplementary Fig. 9 in the revised manuscript.

Figure R3.3 | Conductance map of twist bilayer graphene shows the splitting of the neutrality lines. a, Map of tunnelling conductance of twisted bilayer graphene as a function

$V_t + V_b$ and $V_t - V_b$. The black, red and blue dashed lines mark the sections of constant electric field with $V_t - V_b$ of 1 V, 0 and -1 V, respectively. **b**, The corresponding transport curves showing the splitting of the neutrality line.

4. There should be mentions for scale bar sizes in Supplementary Fig.2 b, c and Fig. 6 a. Thank you for noticing this. The scale bar sizes have been added in the revised manuscript.

Response to comment from Reviewer #4

The manuscript presents an elegant and conceptually innovative study of electrochemical hydrogenation and proton transport in twisted bilayer graphene (TBG) under dual ionic gating. Building on the authors' recent paper in Nature demonstrating decoupled control of proton transport and hydrogenation in monolayer graphene, this work extends the paradigm to twisted bilayers, exploiting twist-induced electronic decoupling to achieve layer-selective hydrogenation and logic operations within a single device. This represents a significant advance: by leveraging the independent carrier densities emerging from the two-layer electronic systems of large-angle TBG, the authors introduce a new type of electrode–electrolyte interface in which electrochemical processes can be spatially and electronically partitioned at room temperature. The demonstration of configurable NOT, NOR, NAND, and XOR logic elements—arising from a single material platform and controlled solely via gate voltages—is very nice, and opens promising avenues for ion–electron hybrid information processing.

The experimental methodology is sound, and the characterization (conductance mapping, tunneling measurements, Raman spectroscopy, and DFT calculations) provides a largely coherent picture of the underlying processes. The work is also clearly novel relative to the group's earlier monolayer study: the ability to target hydrogenation to an individual graphene layer at fixed overall charge density, and to use this selectivity as a functional device degree of freedom, is new, nontrivial, and likely to be of broad interest to the community.

That said, two aspects of the study would benefit from further clarification and tempering of claims.

We are grateful for this encouraging evaluation of our manuscript and for the fair and focused review of our results.

1. Lack of systematic controls on untwisted (AB-stacked) bilayer graphene.

The manuscript includes only a minimal control measurement on AB-stacked bilayers—namely, a conductance map showing a single neutrality line. While this does verify the expected absence of layer decoupling, it does not test whether hydrogenation, proton transport, or logic functionality differ qualitatively from the twisted case. Because the main claims rely on the emergence of layer-selective behavior due to twist-induced electronic isolation, a more complete comparison with untwisted bilayers would significantly strengthen the conclusions. Even basic measurements—such as whether AB bilayers undergo single-layer or simultaneous dual-layer hydrogenation under equivalent fields, or whether proton transport is observable at all—would help establish that the observed phenomena are

uniquely rooted in twist-angle physics rather than more general features of bilayer graphene under high electric fields.

We agree with the Reviewer's comment. In the revised manuscript, we have now included the requested control measurements on AB-stacked bilayers. As expected, the behaviour of AB bilayers is qualitatively different from that of twisted bilayers.

First, natural AB bilayers are impermeable to protons, even under large fields. We demonstrated this in our previous work (ref. 2, Fig. 1b), and we now include equivalent measurements in the revised manuscript (Supplementary Fig. 7e), confirming the absence of proton transport in AB bilayers. Second, because the two graphene layers in AB stacking are electronically coupled, it is not possible to induce layer-selective conductor–insulator transitions. When a sufficiently large potential is applied to one side, the upper surface of the bilayer can be hydrogenated, which is reflected by a conductance drop at the corresponding potential (Supplementary Fig. 7c). However, the bilayer remains overall conductive. Only when a large potential is subsequently applied to the opposite side, does the system undergo a full insulating transition (Supplementary Fig. 7d). These control measurements show that twist-induced electronic decoupling is essential for enabling both layer-selective hydrogenation and logic functionality.

2. Interpretation of the AA-stacked regions as the exclusive proton-transport pathway. The manuscript repeatedly attributes proton permeation primarily to AA regions of the Moiré superlattice. Although this interpretation is physically plausible and supported by DFT barrier calculations, it remains hypothetical within the context of the experimental data presented. The proton-current measurements are spatially unresolved, and no experiment directly evidences transport localization to AA regions or correlates permeation with twist-angle-dependent AA area fraction. I recommend that the authors moderate the certainty of these claims and clarify that the AA contribution, while consistent with theoretical expectations, is not experimentally demonstrated. Future work—such as twist-angle dependence studies, spatially patterned gating, or correlated imaging—could substantiate this point.

We agree with the Reviewer. We have moderated this claim in the abstract and main text.

In summary, this is a strong manuscript that presents noteworthy advances in electrochemical control of two-dimensional heterostructures. With a modest softening of the AA-region interpretation and consideration of additional bilayer-control measurements, the paper will provide a clearer and more rigorously supported account of an important new electrochemical phenomenon in twisted bilayer graphene. I find the work to be of high quality and recommend publication after these points are addressed.

We are truly grateful to the Reviewer for their strong support of our work.

Response to comments from Reviewer #1

The authors have provided a comprehensive and persuasive response to the initial review, significantly strengthening the manuscript through additional experiments, characterization, and discussion. This study remains a highly compelling demonstration of a novel electrode-electrolyte interface concept using twisted bilayer graphene. All major concerns raised have been effectively addressed:

1. The crucial question regarding the coherent bilayer nature of the interface has been conclusively resolved with the addition of atomic-resolution STEM imaging, which directly reveals the moiré pattern. This evidence, combined with the existing transport, Raman, and proton-blocking data, solidly validates the twisted bilayer system.
 2. The practical applicability of the memory function is now strongly supported by the new cyclability data showing >1000 reproducible hydrogenation-dehydrogenation cycles with stable performance, moving beyond the initial impressive state retention.
 3. The logic gate demonstrations are now underpinned by quantitative performance benchmarks, including ON/OFF ratios and cycle variability under standardized conditions. The discussion of switching speed, supported by prior work, adds crucial context for assessing potential applications.
 4. The manuscript now more precisely articulates the conceptual leap enabled by the decoupled bilayer architecture and provides a concrete outlook on extending this approach to other 2D heterostructures.
- We thank the Reviewer for their support of our work.

Response to comments from Reviewer #2

The authors have revised the manuscript and addressed most of the reviewer comments. However, the key claims of the manuscript are still not supported by sufficient experimental evidence, and therefore I cannot recommend publication of the current version.

The thickness of the hBN spacer is only 20 nm, while the width of the hBN aperture appears to be on the order of ~10 μm . Under this geometry, it is highly plausible that the top and bottom graphene layers come into physical contact at the center of the aperture. If such physical contact occurs, the electron transport between the top and bottom graphene layers should not be described as tunneling current. Therefore, the authors must provide direct experimental evidence demonstrating that the top and bottom graphene layers are not in physical contact within the aperture region. Otherwise, the use of the term “tunneling current” is not justified and should be avoided throughout the manuscript.

We believe there is a misunderstanding. The top and bottom graphene layers are intentionally in direct contact across the entire aperture, as the hBN spacer is fully etched in this region. This is essential for device operation. Although the layers are in physical contact, they are electronically decoupled due to the large twist angle. As a result, charge transport across the aperture occurs via electron tunnelling across the interlayer vacuum gap between the two graphene sheets, rather than through hBN. Describing this process as tunnelling is established in the literature for twisted graphene systems. See, for example, P. Rickhaus *et al.*, *Sci. Adv.* 2020, 6, eaay8409. [science.org/doi/10.1126/sciadv.aay8409](https://doi.org/10.1126/sciadv.aay8409); A. Mrenca-Kolasinska *et al.*, *2D Mater.* 2022, 9, 025013. iopscience.iop.org/article/10.1088/2053-1583/ac5536.

Response to comments from Reviewer #3

The authors have adequately addressed the reviewers' concerns. Therefore, I recommend this manuscript for publication in *Nature Communications*.

We are grateful to the Reviewer for this recommendation.

Response to comments from Reviewer #4

The response to referees is substantive, and I think the manuscript is acceptable for publication.

We thank the Reviewer for their support of our manuscript.